# Left to Their Own Devices? A Mixed Methods Study Exploring the Impacts of Smartphone Use on Children’s Outdoor Experiences

**DOI:** 10.3390/ijerph18063115

**Published:** 2021-03-18

**Authors:** Jonas Vestergaard Nielsen, Jan Arvidsen

**Affiliations:** Active Living, Department of Sports Science and Clinical Biomechanics, University of Southern Denmark, 5230 Odense, Denmark; Jarvidsen@health.sdu.dk

**Keywords:** smartphones, mobile technology, outdoor experiences, nature, children, mixed methods, interviews, questionnaire

## Abstract

The growing use of smartphones has been pointed out as one of the main reasons for the decrease in children’s outdoor time. However, there is still a gap in our understanding of how smartphone use affects children’s outdoor experiences and activities. The aim of the study is to explore children’s dependency on their smartphones, what smartphone functions children use when outdoors and how smartphone use affects children’s outdoor experiences. The study uses a mixed methods design which implements interviews with a small sample of children (*N* = 34) in order to help develop a questionnaire for a larger sample (*N* = 1148). Both datasets are included in the analysis with a complimentary perspective. The results suggest that children are highly dependent on having their smartphones available as an integrated part of their lives. However, smartphones also create favorable conditions for rich and valuable outdoor lives by expanding children’s and parents’ sense of security, children’s outdoor sociality, and children’s opportunities to mold their outdoor experiences. We stress that children’s passion for the digital world needs to be reconsidered as not ‘all bad’, but more as a condition in modern children’s lives and an asset to embrace in future strategies for actively engaging children in outdoor activities.

## 1. Introduction

Being outdoors has shown positive health impacts, increased well-being [1,2,3] while also stimulating motor skill development and movement in children [4,5,6]. However, while hanging out with friends and technology-based activities are popular, children do not interact with nature in the same way and with the same frequency as previous generations [7,8,9]. Especially the last decades, increase in smartphone use is pointed out as one of the main reasons that children spend less time outdoors and show an increasingly inactive and sedentary behavior [10,11,12]. Some studies have demonstrated how smartphones, as part of an organized outdoor activity, can motivate children’s active play and physical activity [13,14]. Still, electronically mediated games, albeit playful, is generally perceived to be inferior to analogous play with regards to supporting a healthy childhood and mental and physical development [15]. Nonetheless, smartphones are a natural part of our modern tech-based society that inevitably has weaved itself into the fabric of children’s everyday lives [11,16,17,18]. This has prompted children to develop a strong set of digital skills and to constantly explore the many opportunities available online [19,20]. The embedding of smartphones in children’s lives has resulted in the use of smartphones as a common and intuitive way of interacting with other people, places and objects. Furthermore, there are indications that parents are more inclined to support children’s independent mobility when children have a phone [21,22]. This suggests that smartphones not only provide access to new places and possibilities of being physically active, but also mediate the process of encouraging children to go outdoors and engage with natural environments [10,23,24]. Notwithstanding the rapidly growing body of research within the field, there is still a gap in our understanding of how the use of smartphones affects children’s interactions with and experiences of the socio-material world outdoors. Thus, the aim of the current study is to explore three research questions: (i) How dependent are children on their smartphones? (ii) What smartphone functions do children use in their outdoor lives? and (iii) How do smartphone use affects children’s outdoor experiences? This will help shed a light on the relationship between children’s smartphone use and engagement with and experiences of their everyday outdoor environments when they are left to their own devices, and hence strengthen the needed knowledge for future strategies for actively engaging children in outdoor activities with or through their smartphones [10,18,23].

## 2. Materials and Methods

By implementing interviews with a small sample of children in order to help develop a questionnaire for a larger sample, the study employs an iteration mixed methods design [25]. In the present study, the quantitative analysis delivers the key findings. However, the two methods are integrated through a joint analysis with a complimentary purpose [26] in order to broaden the overall understanding of how smartphone use affects children’s outdoor experiences. All children who participated in this study were 5th to 8th graders (between 11 and 15 years old) and recruited through a sample of Danish schools. Data was generated during schooltime. In Denmark, school attendance is mandatory for children aged 6–16 years. Although data was generated during schooltime, both the questionnaire and the interviews solely concerned the children’s outdoor experiences in their leisure time.

Socioeconomic status (SES) has shown to be important both for children’s use of smartphones, and for their outdoor recreation [8,20,27]. In the present study SES is assessed through parents’ yearly household income. Furthermore, the building density of an area (access and availability of nature) is also assumed to affect the natural use of the age group [28,29]. In the present study, building density is assessed based on whether the schools were located in rural, suburban or urban areas. The selection of schools is based on these two overall conditions.

### 2.1. Interviews

The main goal of the interviews was to create an understanding of how the target group use their smartphones and how they think about their use when they are outdoors in their leisure time. This was done through three main themes regarding children’s use of (i) smartphones, (ii) outdoor areas and (iii) smartphones when outdoors. For the full interview guide, please refer to Appendix A.

Three schools in the region of southern Denmark assisted in the recruitment of children and retrieval of parental consent. In total, 35 children from 5th to 8th grade were interviewed. The three schools are located in areas with different building density (one urban, one suburban and one rural). Two of the schools had medium SES and one had low SES. Focus group interviews were conducted in groups of 2–3 children from the same grade and school in an effort to level the power inequality between the researcher and the children [30,31] and to yield a greater depth and breadth in responses than would be possible in individual interviews [32,33,34]. The interview groups were formed through discussions with the children and their class teachers in order to ensure that the interviewees felt safe and comfortable. Doing interviews in small groups may have biased the children’s answers. Judging, however, from the positive interplay between the children and their general open-heartedness during the interviews, the benefits of doing group interviews by far outweighed the pitfall of obtaining biased reports from the children. A semi-structured interview guide was used to facilitate the interviews, which were audio recorded and subsequently transcribed. Interviews were collected during wintertime in December 2018.

### 2.2. Questionnaires

In addition to the interviews, a questionnaire was constructed. The development of the questionnaire was informed by findings from the interviews, and by previous questionnaires investigating children’s screen time and media habits [35,36]. The aim of the questionnaire was to investigate 5th–8th grade children’s general and outdoor use of smartphones in their leisure time. Like the in interviews, the questionnaire also divided into three main themes regarding children’s use of (i) smartphones, (ii) outdoor areas and (iii) smartphones when outdoors. Children were mainly presented with questions containing yes/no answers (e.g., ‘Do you bring your smartphone when you’re outside?’) or by picking from a list (e.g., ‘Which of the following features do you use smartphone for when outside?’). The questionnaire took approximately 15 min to complete. Invitations for schools to participate in the questionnaire was sent out to 188 public schools from 25 municipalities (out of the 98 Danish municipalities) across different parts of Denmark and with a variation in SES and rural, suburban and urban location. Based on the invitations, 11 schools chose to participate (*N* = 2647 children from 5th to 8th grade). The participating schools were located in different parts of Denmark and in areas with different building density (five urban, two suburban and four rural). Both schools with high, medium and low socioeconomic backgrounds participated, yet with a higher representation of medium SES. The questionnaires were electronically distributed to the children by their teacher during the school day. Questionnaire data was collected during springtime in May 2019.

### 2.3. Data Analysis

Interviews were analyzed using qualitative content analysis [37]. Ultimately, this entails a systematic description of the empirical data through coding [37]. The interviews were initially transcribed, and the transcript was pseudonymized. Subsequently, the first author thoroughly familiarized himself with the data prior to coding the material. This familiarization was done by repeatedly listening and reading the material while taking notes. Subsequently, the first author trial-coded a large portion of the interviews according to a coding-frame. The coding-frame was constructed to embrace a set of categories representing the main themes of the interview guide; children’s use of (i) smartphones, (ii) outdoor areas and (iii) smartphones when outdoors. Through the data-driven trial-coding, subcategories inductively emerged and were used to adjust and refine the coding framework. The results from the trial-coding were assessed by the second author through peer-debriefing, systematically and critically examining the consistency and adequacy of the coding [38], following a discussion among the authors in order to reach consensus. Subsequently, all data that had been used in the trial-coding was recoded with the adjusted coding-frame along with the rest of the interview data. Based on this final coding, main findings informed the development of the questionnaire through explanatory text and specific items based on e.g., the social media applications that children use (e.g., most used Snapchat and TikTok, while not having a Facebook or Twitter account); how their pets were viewed as a social companion when outdoors; how their smartphones instill a sense of security when outdoors; the importance of being able to play music on their smartphone and the general importance of having social relations with their peers when outdoors.

Questionnaire data was downloaded in a csv-format from the electronic survey program SurveyXact and imported into Stata version 16 to perform analysis. Initially, the dataset was cleaned for any obvious typing mistakes. Descriptive analyses were performed to examine the proportion of children owning a smartphone, the dependence on smartphone use and how smartphone use influenced their outdoor experiences. Pearson chi-square analysis was employed to test for difference at a 0.05 significant level across gender (male vs. female) and grades (5th to 8th) with regard to the children’s smartphone use outdoors as well as how smartphone use affected their outdoor experience. This was due to both gender and grade being pointed out to influence children’s use of the outdoors [8,9] as well as both frequency in use and functions of their smartphone [20,21]. 

Subsequent to the questionnaire analysis, the interviews were revisited in order to integrate the two sets of data. In this final step of the analysis, the first author thoroughly read the coded interview material and selected main quotations relating to the children’s use of smartphones outdoors, and how this use influenced their outdoor experiences. This was in order to constitute the complimentary perspective on the findings in the survey [26]. 

### 2.4. Ethics

Written informed consent, also containing consent for publication, was obtained from the parents of the participating children prior to the interviews [39]. The interviewer shortly informed the children about the study at the beginning of each interview and obtained an informal consent from the children to participate [40,41]. Further, the interviewer emphasized that the children were free to withdraw from the interview at any time and without explanation [39]. For further information please refer to Appendix B containing the information letter and the consent form. 

The questionnaire was anonymous and did not collect personally identifiable information from the children such as name, address or civil registration data. The schools distributed an information letter to the parents to inform them and their children about the study. Teachers were also given a short manuscript that they were advised to read out before introducing the questionnaire to the class. A passively informed consent procedure was used, automatically including the children unless the children or their parents actively withdrew consent. It was made easy for both children and parents to withdraw from the study without providing any explanation. To withdraw from the study, children or parents had to inform the teacher, the school or the research-team through the given contact-information in the information letter. This procedure is in accordance with Danish regulations in anonymous and low-risk research and been found to be ethically appropriate for the target group [42]. The study was approved by the Research and Innovation Organization (RIO), University of Southern Denmark (10.327), and conducted according to the World Medical Association Declaration of Helsinki.

## 3. Results

Across the 11 schools 1148 5th–8th grade children answered the questionnaire, which resulted in a response rate of 43%. An equal number of boys and girls completed the questionnaire, but the data showed a slight over-representation of 5th and 6th grade children (Table 1). Interview informants were equally distributed among 5th–8th graders and had a small over-representation of girls (59%) (Table 1). Both in the questionnaire and the interviews almost all children reported to have a smartphone (99% and 100%, respectively). 

### 3.1. How Dependent Are Children on Their Smartphones?

Questionnaire results show that 72% of the children always have their smartphone on them, and this is the case for 63% when being outdoors (Table 2). 

The results of the chi-square analysis comparing the percentage for different subgroups show that there is no difference between boys and girls, but that there is a significant difference between grades and that older children have their smartphone more available than younger children (Table 2). This difference between age groups is also supported by the interviews, as especially 8th graders express that their smartphones are an integral part of their daily lives and social interactions. A girl and a boy illustrate their dependence on their smartphone:
Girl: My smartphone is really important, you know. It’s my life!Boy: I agree, it’s also really important to me … mostly because of keeping up with my social media and being able to be in contact with the world …[8th grade children]

Interviews also show that smartphones are considered a less essential part of the 5th graders’ everyday lives. These children generally express less dependency on their phones, and many voice that it is not that important to carry their smartphone with them all the time. As two 5th graders express, when asked how they would react if they have forgotten their smartphone at home:
Interviewer: So, it’s not like if you’ve forgotten it [the smartphone] at home, you’re going to be like, oh no!Girl: It’s not a problem. I’ll still get through the day … I usually just use it if I’m bored or something, or to call someone.Boy: I agree, I wouldn’t panic … I mostly use it for social media and sometimes to play games if I’m bored.[5th grade children]

### 3.2. What Smartphone Functions Do Children Use in Their Outdoor Lives?

Table 3 shows questionnaire results regarding the smartphone functions that the children use outdoors. The main smartphone function used when outdoors is listening to music (66%). There is both a significant difference in the use of music across gender (<0.01) and grade (<0.01), showing that girls listen more to music on their phones than boys and older children listen more than younger (Table 3). 

These results resonate with the findings in the interviews, where listening to music is highlighted as an important smartphone-function, and especially something that is tied to outdoor use:
Boy: I don’t listen to music much when I’m inside the house.Girl: No, me neither … it’s more fun to listen to music outdoors, when you can watch the surroundings and walk around instead of just sitting inside, staring into the wall and listening to music …Interviewer: Would you also go for a walk without listening to music?Girl: No. That would be boring.[5th grade children]

The second most frequently used smartphone function used outdoors is being in contact with parents (62%) followed by being in contact with friends (57%), taking pictures and video (53%) and using social media platforms (47%). The analysis shows a significant difference across grades regarding the use of smartphones to be in contact with friends (<0.01) and to be present on social media platforms (<0.01) (Table 3). Furthermore, boys tend to watch videos and play games significantly more than girls (<0.01) when outdoors. Results also show a significant difference across grades regarding the use of smartphones to find information when outdoors, with 8th graders being the ones who most frequently report this use.

Regarding contact with parents, there were only small differences across genders and grades, and the frequencies were all rather high (59–65%). The interview findings support these results, as all informants report that the ability to get in contact with their parents is a key feature of having a phone. Furthermore, the majority also reveal that they initially were given a smartphone by their parents, so they were able to reach them. As two 6th graders explain, this is partially due to the ability to satisfy parents’ need for peace of mind regarding the children’s whereabouts and possible change of plans:
Girl: Well, I’d like to bring my phone to school so I can call people if plans change after school … call my mom or something …Boy: Yeah, if you don’t want to go straight home after school … Or if your bike has a flat tire …Girl: I got my phone from my parents mostly because I should be able to call my mom …Boy: Me too, so you can get a hold of people … of my parents, if I need them … also, they are able get a hold of me[6th grade children]

This indicates that the mere ownership of a smartphone gives the children greater freedom to move around independently. Accordingly, Table 3 shows that 23% of the children use their smartphones to help find their way when outdoors. Results show a significant difference across grades (<0.01), having older children reporting this more frequently than younger children, which could imply that the older children tend to visit more unfamiliar outdoor areas. These results could imply that the ownership of a smartphone influences older children’s independent mobility in two ways; both through their parents’ increased sense of security and through the ability to navigate unfamiliar areas.

Table 3 also shows that three out of the five most used smartphone functions relate to being in contact with other people, which clearly illustrates that phones affect children’s social lives. Findings from the interviews resonate with this result by highlighting how smartphones increase children’s outdoor sociality. As the children explain below, smartphones enable social interaction across physical space (e.g., when being alone) and also across time, as they are able to create or catch up on messages/posts:
Boy: Sometimes when I’m walking my dog, I use my phone.Girl: Right, me to … then you can just check Instagram and Snapchat and see if you’ve gotten any messages.Boy: Exactly …Interviewer: So, it’s something that you just do all the time?Girl: Yes, it’s actually all the time … I mostly listen to music when I’m walking alone outside, but if I get a message from a friend, I can just answer it and keep in touch like that. No matter where I am …[8th grade child]
Girl: If you’re outside … walking or whatever … and you get bored, you just Snap [use the application Snapchat] while you’re doing it.Boy: Right, so you can stay in touch or include your friends when they can’t be there, and then catch up later[6th grade child]

### 3.3. How Do Smartphones Affect Children’s Outdoor Experiences?

Table 4 shows questionnaire results on how the use of smartphones has mediated the children’s outdoor experiences. 

Enhanced feelings of being safe is the most reported benefit of having a smartphone when outdoors (41%). Especially girls report to feel safer outdoors when they bring their smartphone (<0.01). The sense of security was also a salient issue in the interviews, as the children reported that their smartphones reduced their concern about accidents and unfortunate occurrences. This focus on security indicates that the smartphones not only dim parents’ sense of security (Section 3.2 and Table 3), but also help control the children’s what-if mentality, ultimately enhancing their independent freedom to roam and do things outdoors:
Girl: I horseback ride a lot, and my parents said that I can’t go alone unless I have my smartphone with me and are able to call them … So, I’m a little anxious if I don’t have it [the smartphone] with me out there, in case something happens … because I’ve tried that before, where something happened, and then it’s nice that you can call someone …Boy: Yeah, I know what you mean, if you are somewhere, like outside, then it is great to have your smartphone on you, because then you can write your parents … You can get in touch with someone if there’s an emergency.[7th grade children]

The second most reported benefit of using a smartphone outdoors, is the experience of being more entertained (31%) followed by being more flexible and able to act on their own initiative (26%). Analysis entail that boys, more often than girls, report that their smartphone makes them feel more entertained and able to act on their own initiative (<0.01). In total, 23% of the children report that having their smartphone has not affected their outdoor experiences, of which 8th graders report this significantly more than 5th graders (<0.01). 

The interviews show that smartphones empower children to mold the atmosphere of an outdoor area, and thereby making their outdoor experiences more entertaining or cozy. In particular, listening to music seems to have an evocative impact on the children’s experiences of the ambience of an outdoor space. As a 5th grader and the two 7th graders explain:
Boy: When I ride my bike to school it can get very boring … It’s also dark now [in the winter season] … There isn’t really anything you can look at, there isn’t really anything going on, so I listen to some music … Just to make things a bit nicer.[5th grade child]
Interviewer: Are there any specific places outside where you use your phone?Boy1: Sitting at a train station … definitely …Boy2: Yeah, or sometimes when I go to school, I am listening to music.Boy1: Or when you’re going up near the after-school classes, you’re looking down into your phone a lot!Boy2: I agree, it’s not a very exciting setting up there.Boy1: No, it’s just huge gray buildings.[7th grade children]

## 4. Discussion

The aim of the current study is to shed a light on how smartphone use affects children’s engagement in, conditions for and experiences of their outdoor activities. This is done by addressing three main questions: (i) How dependent are children on their smartphones? (ii) What smartphone functions do children use in their outdoor lives? and (iii) How does smartphone use affects children’s outdoor experiences? The results show that children are highly dependent on their smartphones, both when indoors and outdoors. Almost all children own a smartphone (99%) and the majority feel it is important to have it available outdoors. This suggests that phones are an integral part of children’s outdoor lives; unlike other types of screens, such as TV, computers, game consoles, and tablets that are much more connected to indoor use [22]. The results further indicate that this dependence relates to three main aspects of how the outdoor use of smartphones expands (i) children’s and parents’ sense of security when children are independently roaming outdoors, (ii) children’s ability to socialize outdoors, (iii) children’s opportunity to mold their outdoor experiences. As a note, the questionnaire data shows that 23% of the children report that their smartphones do not affect their outdoor experiences. This was especially the case for 8th graders. However, 77% of the 8th graders still reply that they always have their smartphone with them when outdoors. 

### 4.1. Smartphones Increase a Sense of Security When Outdoors

Today’s children generally have an increasing consumption of smart technology in their everyday lives [11,12]. Accordingly, a large European study showed that Danish children are among the youngest (11.1 years) owners of a smartphone, and most often have a social media account [20]. The results of the present study highlight the importance of bringing the smartphone outside, as many find the use has made it more fun and added new opportunities to go places or do activities that otherwise would not have been possible (e.g., allowed by their parents). Furthermore, the present study shows that being able to get in touch with parents and friends when moving around outdoors increases the children’s feeling of security. This both includes security in being able to get in touch with someone (adults) in case of accidents, and more so, a security in being able to inform parents if new opportunities arise and there is a need to change plans during the day. Accordingly, several of the interview informants highlight that the reason they were given a smartphone was to be able to get in touch with the parents. The results resonate well with several other studies showing that owning a smartphone reduces the parents’ perceived risk of children independently transporting themselves around outdoors [21,44]. Other studies have also pointed out that parents allow their children to move outdoors on their own and to extend their range of motion to a higher degree—and even the possibility of moving in otherwise forbidden areas—when they bring their phone [22,44]. In keeping with these results, smartphones can be viewed as an important asset in children’s outdoor lives, because it provides a safety line for parents that increases the opportunities for children’s independent outdoor behaviors.

### 4.2. Smartphones Empower Children to Socialize Outdoors 

The results of the present study also emphasize the importance of smartphones as an asset to remain social when being outdoors. In general, the increased opportunities for using a phone and attending social media accounts wherever they are, have had a major impact on the everyday lives of children [16,21]. For example, Machackova and Olafsson found that 9–12-year-olds considered smartphones and social media to be one of the most important tools to uphold their social needs and a common way of finding new friends [45]. In the present study, 57% of the questionnaire respondents use their smartphone outdoors to stay in contact with their friends and 47% used social media. This was supported by the majority of children in the interviews who expressed a desire to use their social media accounts when they were outside. The need to maintain social relations can be seen in the light of a perspective that several studies highlight: the fact that the smartphone is an important social tool in children’s lives also implies that they may experience social exclusion if they do not have a smartphone, or are separated from it [21,46,47]. This also applies to children’s outdoor lives [48]. The interviews show that social relationships and friends are very important elements in the everyday lives of the target group—also outdoors. This is supported by Halldén, who points out that our outdoor behavior is intertwined in varies motives and relations, thus, we often do not go outside to be outdoors, but because the activity we want to do takes place in the outdoors [15]. The present study supports the point that the activity and the social interactions—rather than the outdoor space itself—are the key driver in children’s outdoor lives. As both children’s outdoor activity and social interaction are mediated by smartphones, we support the notion that such devices may hold promise for supporting children’s outdoor lives [10]. 

### 4.3. Smartphones Empower Children to Mold Outdoor Experiences

Results show that listening to music is the most used smartphone function outdoors. The importance of children’s use of music outside is also documented by others [48] and appears as a powerful path to alter the ambience of outdoor spaces—both when you are alone and with friends. Music has proven to have a therapeutic effect [49] and to be instrumental in helping children to process social situations and improve their well-being [47]. In the interviews, the children express that they are listening to music especially when outdoors and that they instinctively use the music to create a cozy or relaxing atmosphere outdoors. This has also been pointed out by others, illustrating that music can create an experience of a safe and relaxing refuge when children move about outdoors [48]. Children’s all-encompassing absorption in the digital world has blurred the distinctions between the physical and the virtual domains of childhood, which prompt a need to reconsider our assumptions of children’s spatial experiences [47,48,50]. Consequently, smartphone use may have the capacity to alter the ambience of children’s spaces and thereby support valuable experiences, which may promote children’s affinity for the outdoors. Thus, the present study supports the claim that smartphones may be conducive to children’s engagement with nature through appropriate technology-centered activities in outdoor settings [10,23,51,52].

### 4.4. Strengths and Limitations

A strength of this study is the mixed methods design, integrating both questionnaire and interviews to explore how smartphones affect children’s conditions for and experiences of their outdoor activities. Combining quantitative and qualitative methods with a complementing purpose, ensures richer and more credible data to support the understanding of the interactions between the children, their smartphones and the outdoors [53,54]. Additionally, this design was strengthened by having a fairly large number of respondents in both the interviews (*N* = 34) and the questionnaires (*N* = 1148) with variations in sizes, SES and geographic location. Still, it must be duly noted that questionnaires had a response rate of merely 43%. The questionnaire was distributed to whole classes by local teachers during schooltime. Thus, the low response rate did not reflect children choosing not to participate, rather it speaks of teachers not prioritizing the time to present the questionnaire. However, this entails that despite the low response rate, the questionnaire still managed to capture the diversity of students in each classroom participating in the study [55,56].

Furthermore, in Denmark there are large seasonal variations that are assumed to affect the duration and activities that children do when outdoors [8,57]. In the present study interviews were collected during wintertime whereas the questionnaire was collected during springtime. Through the combined analysis, identifying mechanisms across both interviews and questionnaires, the findings are expected to represent both winter and spring seasons. Still, smartphone use outdoors and children’s experiences hereof might vary across seasons, which may compromise the validity of the findings.

## 5. Conclusions

The study indicates that children are highly dependent on having their smartphones available, and this dependency seems to be increasing with age. However, when left to their own devices, smartphones do not hinder children’s outdoor lives, but may—on the contrary—create favorable conditions for rich and valuable outdoor lives by expanding (i) children’s and parents’ sense of security, (ii) children’s outdoor sociality, and (iii) children’s opportunity to mold their outdoor experiences. The study especially highlights how children’s smartphones add new opportunities to go places or do activities that otherwise would not have been possible, while also changing the ambience of the outdoors for the better. Furthermore, smartphones enable continuous social interactions that are a key driver in children’s outdoor lives. Hereby we suggest that children’s affinity for the digital world should be reconsidered: it is not ‘all bad’ but should instead be regarded as a condition in modern children’s lives and an asset to embrace in future strategies of actively engaging children in the outdoors.

## Figures and Tables

**Table 1 ijerph-18-03115-t001:** Overview of the children participating in the interviews and questionnaires.

		Questionnaire Respondents (%)	Interview Respondents (%)
*N*		1148	35
Gender	Boys	576 (50%)	14 (41%)
	Girls	572 (50%)	20 (59%)
Grade	5th grade	329 (29%)	9 (26%)
	6th grade	330 (29%)	9 (26%)
	7th grade	260 (23%)	8 (24%)
	8th grade	229 (20%)	8 (24%)
SES (in 1000 DKK) ^a^	Under average (<250)	389 (34%)	12 (34%)
	Average (250–350)	662 (58%)	23 (66%)
	Above average (>350)	67 (6%)	-
Building density ^b^	Urban	743 (65%)	12 (33%)
	Suburban	146 (13%)	11 (32%)
	Rural	259 (23%)	11 (32%)
Smartphone relationship	Owns a smartphone	1136 (99%)	34 (100%)

^a^ SES (Socioeconomic Status) is presented through parents yearly disposable household income from Statistics Denmark [43]. ^b^ Building density was based on the location of the school.

**Table 2 ijerph-18-03115-t002:** Questionnaire response on the children’s connectedness to their smartphone, in %.

	Total	Gender		Grade			
		Boy	Girl	5	6	7	8
Always have my smartphone with me	72%	71%	74%	56% *	71% *	83% *	80% *
Always have my smartphone with me when outdoors	63%	61%	65%	46% *	63% *	70% *	77% *

* Chi-square analysis: *p* value < 0.01.

**Table 3 ijerph-18-03115-t003:** Smartphone functions that the children use outdoor across gender and grade, in %.

Smartphone Functions Used Outdoor	Total	Gender		Grade			
		Boy	Girl	5	6	7	8
Listening to music	66%	61% **	69% **	56% **	64% **	66% **	80% **
Being in contact with parents	62%	60%	64%	65%	63%	59%	62%
Being in contact with friends	57%	57%	56%	44% **	59% **	62% **	64% **
Taking pictures or video	53%	41% **	65% **	48%	57% *	54%	56%
Using social media	47%	44%	49%	27% **	45% **	56% **	64% **
Use as a clock or alarm	43%	41%	44%	35% *	46% *	45% *	46% *
Help finding the way	26%	25%	27%	14% **	23% **	34% **	36% **
Watch videos	18%	21% **	14% **	15%	16%	17%	20%
Playing games	17%	22% **	12% **	15%	18%	18%	14%
Finding information	17%	20%	15%	13% **	16% **	17% **	25% **
Tracking physical activity and exercise	13%	11%	14%	11%	12%	13%	15%

* Chi-square analysis: *p* value < 0.05. ** Chi-square analysis: *p* value < 0.01.

**Table 4 ijerph-18-03115-t004:** Children’s response on how the use of smartphone has affected their outdoor experience across gender and grade, in %.

Using My Smartphone Outdoor Resulted in Me Feeling:	Total	Gender		Grade			
		Boy	Girl	5	6	7	8
Safer	41%	36% **	42% **	40%	47%	37%	40%
More entertained	31%	36% **	25% **	28%	30%	33%	35%
More flexible and able to act on my own initiative	26%	30% **	20% **	18% **	29% **	31% **	26% **
Using my smartphone has not affected my outdoor experience	23%	25%	20%	18% **	22% **	21% **	33% **
Less observant of my surroundings	18%	15% *	19% *	15% *	14% *	21% *	23% *
Less in contact with the people I am with	15%	12% *	17% *	13%	12%	17%	20%
More educated	13%	17% **	9% **	10%	14%	16% *	13%
More bored	8%	7%	8%	11%	7%	8%	6%

* Chi-square analysis: *p* value < 0.05. ** Chi-square analysis: *p* value < 0.01.

## Data Availability

The data presented in this study are available on request from the corresponding author. The data are not publicly available due to legal and privacy issues.

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
