# Peer review of "Left to Their Own Devices? A Mixed Methods Study Exploring the Impacts of Smartphone Use on Children’s Outdoor Experiences"

_ijerph, 2021, doi:10.3390/ijerph18063115_

Round 1
Reviewer 1 Report
- This is an interesting mixed methods study to explore children’s use of smartphone in outdoor. Overall, I found the manuscript well-written. However, there are several major limitations of the study including the use of descriptive statistics, self-report and self-constructed measures, and convenience sampling with low response rate.
- When I first read the title and abstract of the manuscript, I thought “mediate” refers to mediation analysis. I would suggest the authors to consider using an alternative word to replace it throughout the paper.
- Please list the interview questions. Also, it is unclear how the data were coded and analysed. Were the three coding themes (use of smartphones, outdoor areas and smartphones when outdoors) pre-determined by the authors or informed by the data? Please elaborate the coding and analysis procedure.
- As the authors mentioned, one of the strengths of the study is the use of both qualitative and quantitative data. 34 participants were interviewed and I believe there is rich interview data. I wonder what additional qualitative findings could the authors add to the manuscript.
- I do not think regression analysis is appropriate for the study. Chi-square test to examine the pattern of distribution is a better choice.
Reviewer 2 Report
This is an interesting topic of great significance to how we understand children's engagement with technology, social and play behaviors, and use of indoor/outdoor space. The literature review/introduction is well done, and I appreciate the attentive use of iterative mixed-methods methodology. There are a few concerns that need to be addressed for this paper to demonstrate its value and integrity:
1) There are some minor grammar edits that need to be made, likely as a result of translation to English (primarily omissions of prepositions and qualifiers).
2) The word "mediate" is problematic in this descriptive study. The analyses chosen, and the elements highlighted here, do not lend themselves to making statistically supported inferences about the effects of one variable on another. Instead, they can indicate relationships, so I would suggest replacing mediate with "what is the relationship between smartphone usage and children's outdoor experiences." This is more in line with both the goals and the methods/analyses used as this study stands.
3) There needs to be more detail about a) the findings of the qualitative component of the study that generated the questionnaire as this is currently skipped over entirely, and b) the contents of the instrument that was created. It is difficult to assess the reliability and validity of this instrument without knowing how many/what type/what themes etc. were included in the instrument's questions. It is also difficult to assess the developmentally appropriateness of the tool without these specifics. I am also slightly concerned about the use of 2-3 person "interviews" (which I would call focus groups), due to this age group's sensitivities to social dynamics and hesitance to speak openly about their experiences at risk of peer judgment. I would have pushed for a singular person interview for each for this reason.
3) There is not enough detail to rationalize a) the use of regression in this study (since the goal does not seem to be about predicting), and b) why some elements such as gender and grade level are prioritized and reported on in these regressions versus the components of the questionnaire that report on the specific variables of interest - smartphone usage and outdoor experiences. I would need to see either more discussion of why these are not demonstrated, or justification for why the analyses were done in this way.
4) Finally, can you clarify the consent/assent procedures? In the US we would have to gain formal assent from children of these ages regarding their participation in the study, rather than the idea of "passive consent" that is described here. Could you add a sentence or two to clarify whether the child had the opportunity to consent/assent, and not just the parents or teachers?
Round 2
Reviewer 1 Report
The authors have addressed my comments and suggestions adequately. There are very minor typos e.g., "Facebook" in the data analysis section and "Introduction" in the appendix.
Reviewer 2 Report
Thank you for considering and making these revisions. The paper feels much improved and I think will be even better received by the scholarly community.